# Reduction in Acute Bee Paralysis Virus Infection and Mortality in Honey Bees (*Apis mellifera*) by RNA Interference Technology

**DOI:** 10.3390/insects16050453

**Published:** 2025-04-25

**Authors:** Cecilia Ferrufino, Alejandra Scannapieco, Romina María Russo, Fernanda Noemí Gonzalez, Ricardo Salvador, María José Dus Santos

**Affiliations:** 1Instituto de Virología e Innovaciones Tecnológicas (IVIT) INTA-CONICET, Centro de Investigaciones en Ciencias Agronómicas y Veterinarias (CICVyA), Instituto Nacional de Tecnología Agropecuaria (INTA), Hurlingham 1686, Buenos Aires, Argentina; ferrufino.cecilia@inta.gob.ar (C.F.); gonzalez.fernanda@inta.gob.ar (F.N.G.); 2Instituto de Genética Ewald A. Favret INTA-CONICET, Centro de Investigaciones en Ciencias Agronómicas y Veterinarias (CICVyA), Instituto Nacional de Tecnología Agropecuaria (INTA), Grupo Vinculado al IABIMO—Consejo Nacional de Investigaciones Científicas y Técnicas (CONICET), Hurlingham 1686, Buenos Aires, Argentina; scannapieco.alejandra@inta.gob.ar (A.S.); russo.romina@inta.gob.ar (R.M.R.); 3Instituto de Microbiología y Zoología Agrícola (IMyZA), Centro de investigaciones en Ciencias Agronómicas y Veterinarias (CICVyA), Instituto Nacional de Tecnología Agropecuaria (INTA), Hurlingham 1686, Buenos Aires, Argentina; salvador.ricardo@inta.gob.ar

**Keywords:** ABPV, double-stranded RNA, RNA interference technology, honey bees, viral infection, survival

## Abstract

*Aparavirus apisacutum*, the causal agent of acute bee paralysis (ABP), is a highly prevalent pathogen in honey-producing and queen-rearing apiaries. This study evaluated the effect of RNA interference (RNAi) technology, mediated by double-stranded RNA (dsRNA), on the dynamics of ABPV infection in adult honey bees. Fragments of the replicase and VP1 genes were selected for in vitro dsRNA synthesis and subsequently tested in a gene silencing experiment. Oral administration of specific dsRNA significantly reduced viral replication (assessed via RT-qPCR) and improved bee survival. These results demonstrate the potential of dsRNA oral administration to mitigate ABPV infections in adult honey bees, supporting the use of post-transcriptional gene silencing as a promising tool for controlling viral infections in apiaries.

## 1. Introduction

Bee viruses represent a significant threat to honey bee colonies and honey production worldwide [1]. Under optimal nutritional and health conditions, honey bees exhibit both individual and population-level resistance mechanisms that help maintain these viruses at sublethal levels [2]. However, factors such as stress, immunosuppression or nutritional deficiencies, can disrupt this balance, leading to viral infections that manifest as clinical signs of disease and ultimately weaken the colony [3]. The procedural stress and frequent handling inherent in queen-rearing production expose honey bees to conditions that may enhance virus circulation within the colony [4]. In queen rearing, overt infections can severely impact the commercialization and supply of queens to beekeepers, thereby jeopardizing the entire apiary economy.

Ravoet et al. (2015) provided further evidence of vertical virus transmission by detecting the presence of viruses in queen eggs during a sanitary control of a breeding program [5]. Additionally, viral genomes of five viruses, *namely Iflavirus aladeformis* (DWV), *Aparavirus apisacutum* (ABPV), *Triatovirus nigereginacellulae* (BQCV), *Iflavirus sacbroodi* (SBV), and *A. mellifera filamentous virus* (AmFV) were detected in 91% of semen samples from apparently healthy drones used for artificial insemination. This finding suggests that the commercialization of drones may facilitate virus transmission [6]. Moreover, occurrences of ABPV, BQCV, and DWV have been reported in various colony materials (such as tissue and adult feces), workers, nurses, and queens during the breeding process [7]. In an experiment conducted during the queen development stages, viruses were occasionally detected: BQCV was found in all worker samples and in queens from mating nuclei, while DWV was only detected in queen samples [7]. In Argentina, we previously reported the presence and impact of ABPV and BQCV in queen-rearing apiaries. ABPV and BQCV were detected in 62.5% (5/8) and 50% (4/8) of queen pupae samples and co-infections with both viruses were observed in 37.5% (3/8) of queen pupae samples. ABPV was also detected in 33% (2/6) of the royal jelly samples suggesting a putative source of virus that could infect pupae. Most importantly, in queen-rearing apiaries with significant losses in queen cell production, the agents involved were BQCV and ABPV. These results underscore the importance of controlling these viruses in honey bee queens’ production [8]. Particularly, Acute Bee Paralysis Virus (ABPV) is a widespread virus that typically exhibits covert infections and does not cause clinical symptoms. However, when present at high titers, it is associated with significantly increased mortality rates. Affected adult bees often exhibit rapidly progressing paralysis, characterized by trembling, an inability to fly, and gradual darkening of the body. This is often accompanied by hair loss on the thorax and abdomen [9].

Considering the harmful effect of viral infections on honey bee health and survival, the development of effective control technologies is of utmost importance. Gene silencing, mediated by RNA interference (RNAi), is one of the most important defense mechanisms that bees have against pathogens [2]. Focusing on this immune pathway, several authors have conducted gene silencing experiments mediated by double-stranded RNA (dsRNA) to control viral infections. For instance, in 2009, Maori et al. demonstrated a reduction in bee mortality caused by IAPV through dsRNA ingestion [10]. Similarly, in 2012, Desai and colleagues controlled DWV infection in first-stage larvae and in adult bees via oral administration of dsRNA specific to DWV [11]. Moreover, a large-scale field study applying specific dsRNA against IAPV showed that this treatment effectively protected the bee population from viral infection and improved overall colony health [12]. However, to date, no studies have been conducted to assess the effect of RNAi on ABPV infection in adult honey bees.

The aim of this study was to evaluate the effectiveness of RNAi technology, mediated by dsRNA, in reducing ABPV infection and improving the survival of adult honey bees.

## 2. Materials and Methods

### 2.1. Production and Stability of dsRNA

VP1 and replicase gene fragments were amplified by PCR using the primers Fw 5′-TCTGATGATGCTGAAGAGAGAAA-3′) and Rv 5′-AATCATCATTGCCGGCTCTA-3 (Accession number, AF150629), Fw 5′-CTCAAGTTATACGTAAAATAGCTGGAATT-3′ and Rv 5′-AACCAACCTTGCTTCCCTTTA-3′ (Accession number, AF486072), respectively [3,13]. The resulting amplicons (VP1: 500 bp and Replicase: 646 bp) were cloned into the pGEM T-easy vector (Promega Corporation, Madison, WI, USA) according to the manufacturer’s recommendations, and their identities were verified by sequencing. The gene fragments were then individually subcloned into the pL4440 vector using classical cloning techniques [14]. The plasmid pL4440-IAP (IAP fragment gene of Varroa destructor) was used for non-specific dsRNA synthesis [15].

ABPV gene fragments and non-specific gene fragments were used as templates for dsRNA synthesis via in vitro transcription, utilizing the MEGAscript™ RNAi Invitrogen Commercial Kit according to the manufacturer’s instructions. The synthesized dsRNA products were analyzed by electrophoresis on 1% agarose gels and quantified using a Nanodrop2000 spectrophotometer (Thermo Fisher, Waltham, MA, USA). A positive control dsRNA synthesis provided by the kit manufacturer was included. A total of 200 µg of dsRNA was produced for each target.

A stability assay of the synthesized dsRNA was conducted under various conditions over a period of 19 days. For this purpose, 1 µg of dsRNA was aliquoted into 39 tubes (1 μg/tube), with 13 tubes assigned to each condition: diluted in water and stored at 4 °C, diluted in water and stored at 32 °C, diluted in sucrose and stored at 32 °C. On each of the designated days, one sample from each condition was frozen at −80 °C and stored until the end of the assay. The products were subsequently analyzed by electrophoresis on a 1% agarose gel (Appendix A).

### 2.2. Amplification and Purification of Viral Stock

To prepare the infective virus stock, a macerate from ABPV-positive samples from Entre Rios province, Argentina [8], was obtained and quantified by RT-qPCR [16]. In total, 1–5 μL of macerate (containing 10^3^ gc/μL) was inoculated into 44 bee pupae (Appendix A). After 3 days, the pupae were processed in 5 mL of NET buffer (0.1 M NaCl, 0.004 M EDTA, 0.05 M Tris, pH 8.0) and centrifuged at 3645× *g* for 45 min at 4 °C. The supernatant was collected, and subsequent 3 washes with chloroform were performed by centrifuging at 405× *g* for 5 min and collecting the supernatant in each wash. The recovered material was disintegrated by adding 0.5 mL of 6% Sarkosyl for each 2 mL of supernatant, and the mixture was incubated at 4 °C for 15 min. Purification was achieved through continuous sucrose gradients (15 to 41.25%) and ultracentrifuged at 28.000 rpm for 3 h at 4 °C in a Beckman 28.38 rotor [17]. The gradient containing the viral suspension was fractionated into 96-well microplates using a peristaltic pump with a flow rate of 1.12 mL/min. The eluates were collected in order of increasing density and absorbance readings were taken at 260 and 280 nm using a microvolume spectrophotometer (Nanodrop 2000, Thermo Fisher) (Appendix A). The purified viruses were quantified and stored at −80 °C. The absence of DWV, BQCV, IAPV, KBV, CBPV, and SBV was confirmed by RT-qPCR [18].

### 2.3. ABPV Quantification

Pools of 3 honey bees per group were macerated in a mortar with 0.5 mL of PBS pH 7. The complete mixtures were centrifuged at 3645× *g* at 4 °C for 45 min. From the resulting supernatant, 200 µL was utilized for RNA extraction using TIANamp Virus RNA Tiangen^®^ (Changping District, Beijing, China) and following the manufacturer’s recommendations. The remaining supernatant was collected and stored at −80 °C.

The presence and quantification of ABPV were assessed by RT-qPCR following the protocol described by [16]. Briefly, one-step RT-qPCR was performed using iTaq Universal SYBR Green one-step kit (Bio-Rad, Hercules, CA, USA). The reaction mixture contained 1 µL of each primer (10 µM), 0.25 µ iScript RT, 3.75 µL H_2_O and 5 µL of RNA template, in a final volume prepared in white tubes. The thermal cycling profile was as follows: 10 min at 50 °C, 3 min at 95 °C, followed by 40 cycles of 15 s at 95 °C, 1 min at 60 °C and a melting curve from 65 °C to 95 °C increasing 0.5 °C every 5 s. All assays were performed on a CFX96 thermocycler (Biorad).

For quantification, a five-point standard curve was generated with Ct values obtained from known concentrations of a plasmid containing the target sequence. These concentrations were prepared as 10-fold dilutions ranging from 10^2^ to 10^6^ genome copies/µL. Each point on the standard curve was assayed in triplicate. Viral loads (VLs) were expressed as genome copies/µL reaction. The efficiency of the reactions ranged from 95 to 105% (Appendix A).

### 2.4. Bioassays

Adult bees used in the following experiments were obtained from healthy colonies maintained at the research apiary at the Instituto de Genética Ewald A. Favret INTA-CONICET. In total, 30 randonmly selected bees from each colony were collected to determine background virus VLs before the assay.

Frames of emerging brood were removed and placed in an observational colony that was kept for 1–2 days in an incubator at 32 °C and 60% relative humidity. Newly emerged bees from these frames were mixed together and pooled according to the experimental design.

#### 2.4.1. Assay 1—ABPV Infection

To determine the optimal amount of virus dose for subsequent silencing assays, a repeated measures experiment was conducted on adult bees. The experiment involved three treatments, which included two different doses of ABPV (10^3^ gc/µL and 10^4^ gc/µL) and a control group. Each treatment was replicated three times, with each replicate consisting of 50 bees placed in a 3 L glass jar. The bees in each jar were fed ad libitum according to their assigned treatment (Appendix A), as described below.

Newly emerged *A. mellifera* workers were individually marked on the thorax with different colors using a queen bee marker to distinguish bees for survival evaluation (N = 30 in each jar) from those intended for viral load quantification (N = 20 in each jar) (Appendix A). The jars were placed in a breeding chamber (SANYO, Osaka, Japan, Versatile Environmental Test Chamber MLR-350) under controlled conditions of 34 ± 1 °C and 65 ± 5% relative humidity.

On day 0, bees assigned to treatments B and C were fed ad libitum with honey containing 1 mL of ABPV (at a concentration of 10^3^ gc/μL and 10^4^ gc/μL, respectively) per glass jar. Bees in treatment A (the control group) received only honey (Table 1). Three bees from each group were sampled on days 0, 2, 4, 6 and 8 and frozen at −80 °C for subsequent VLs quantification (Figure 1A). From day 2 onwards, all bees were fed with 1 mL of honey and 1 mL of water every 24 h. The number of live bees was recorded daily for 10 days.

#### 2.4.2. Assay 2—Viral Silencing

A second experiment was conducted to evaluate the effect of oral dsRNA administration on VLs and bee survival.

The experiment included five treatments, each with four replicates. In each replicate, 50 bees were placed in a 3 L glass jar and fed ad libitum according to their assigned treatment. Bees were marked with different colors, and the jars were placed in the breeding chamber as described in assay 1.

On day 0, bees in treatment A received only honey (control group). Bees in treatment B received specific dsRNA (at day 0) and virus (at day 3), treatment C received virus at day 3, treatment D received at day 0 specific dsRNA only, and treatment E received non-specific dsRNA (at day 0) + virus (at day 3) (Table 2).

Each replicate (glass jar) was fed ad libitum with honey containing 1 mL of virus suspension containing 10^4^ genomic copies/μL (2 × 10^5^ gc/bee), and the dsRNA dosage per glass jar was 50 μg. The experiment lasted 10 days. Three bees from each group were removed on days 0, 2, 4, 6 and 8, and frozen at −80 °C to subsequent viral load quantification (Figure 2A). All bees were fed with honey and water every 24 h. The number of live bees was recorded daily for 10 days.

### 2.5. Statistical Analyses

The data on VLs were analyzed using a General Linear Model (GLM). The response variable was the logarithm of the virus concentration, while the explanatory variables were time and treatments, both considered as fixed factors. The Shapiro–Wilk test was used to assess the assumption of Normality. Since viral load measurements from the same groups of bees were taken at six different time points, there was a lack of independence between observations. To account for this, a correlation structure (composite symmetry matrix) was applied to the group variable. Additionally, due to the lack of homogeneity of variances across treatments, the variance structure was modeled, incorporating the Identity function (varIdent) into the model.

Survival data were analyzed using a General Linear Model (GLM) to assess differences between treatments and over time treated as fixed factors. The response variable was the rate based on the number of live bees over total bees in each glass jar (N = 30).

Orthogonal contrasts with Bonferroni correction were performed to determine if there were significant differences between treatments at each time point. All analyses were conducted using Infostat Software (2020, http://www.infostat.com.ar (accessed on 29 November 2024)) [19].

## 3. Results

### 3.1. ABPV Infection Kinetics and Bee Survival

The aim of this experiment was to determine the virus dose required to observe the clinical manifestation of ABPV infection and to study the kinetics of viral load (VL) in adult bees. To achieve this, bees were orally infected with two different doses of ABPV, and both VLs and mortality were monitored. On the third day post infection (dpi), an exponential increase in VLs was observed in both ABPV-treated groups (day 2 in Figure 1B). The group that received 10^4^ gc/μL reached a viral load level of 10^9^ gc/µL, while the group treated with 10^3^ gc/μL peaked at 10^5.8^ gc/µL (Figure 1B). In the control group, which started with a basal VL, the viral load increased to 10^4.8^ gc/µL by day 4. VLs remained constant across all treatments from day 4 until the end of the experiment. Regarding survival, a significant decrease was observed on the fourth day post-infection in the group treated with ABPV at 10^4^ gc/µL (Figure 1C). In contrast, the control group maintained a survival rate greater than 85%. Additionally, clinical signs of ABPV infection were observed in the virus-treated groups, including hair loss on the thorax and abdomen, gradual darkening, tremors and an inability to fly (Appendix A).

### 3.2. Viral Silencing Experiment

The effect of oral administration of specific dsRNA to inhibit ABPV infection was evaluated in an in vivo experiment using adult bees. dsRNAs targeting the VP1 and replicase genes were generated as a viral silencing tool.

First, the stability of the dsRNA was assessed in vitro under different conditions over a period of 19 days. It was observed that the dsRNA remains stable in water at 4 °C throughout the experiment. At 32 °C, both in water and sucrose, the dsRNA remained stable until day 18, after which slight signs of degradation were detected (Appendix A).

The experimental design is shown in Figure 2A. On day 4 (1-day post-infection), an exponential increase in average VLs was observed in treatments C (ABPV 10^4^ gc/μL) and E (ABPV 10^4^ gc/μL + non-specific dsRNA). In treatment B, which included the administration of specific dsRNA (ABPV 10^4^ gc/μL + dsRNA), VLs were significantly lower than those in treatments C (*p* = 0.0006) and E (*p* = 0.0306). Significant differences were also observed on day 6 between treatments B and C (*p* = 0.0002) (Figure 2B).

VLs in the control treatments (A and D) slightly increased but remained significantly lower than those in the virus-treated groups (treatments C and E) from day 4 until the end of the experiment.

A significant decline in bee survival was observed in treatments C (ABPV 10^4^ gc/μL) and E (ABPV 10^4^ gc/μL + non-specific dsRNA), starting on day 5 (2 dpi). In treatment B (ABPV 10^4^ gc/μL + specific dsRNA), a slight reduction in survival was noted on day 6. On day 5, survival rates in treatment B were significantly higher compared to treatment E (*p* = 0.0039). By day 6, survival in treatment B was significantly greater than in treatment C and E (*p* = 0.0006 and *p* < 0.0001, respectively). Up to day 7, no significant differences in survival were observed between treatment B and the control treatments (A and D). In the control group (treatment A), 25 to 30 bees remained alive by the end of the experiment, representing a survival rate of 83.3% to 100% (Figure 2C).

The impact of dsRNA administration was quantified, showing an average reduction in VLs of 2.31 to 3.06 log gc/μL (95% CI) at time point 4 (1 dpi), and a reduction of 2.48 to 3.48 log gc/μL (95% CI) at time point 6 (3 dpi).

## 4. Discussion

Some studies have demonstrated the feasibility of controlling viral infections in bees through the oral administration of dsRNA. Maori et al. (2009) reported a reduction in bee mortality caused by IAPV and another large-scale study against that virus confirmed that specific dsRNA treatments were effective in protecting the bee population from IAPV infection [10]. Moreover, Liu et al. (2010) reported successful blocking of CSBV infection using specific dsRNA sequences. DWV infection effects have been also reduced by feeding bees with DWV-specific dsRNA [11,12,20]. Recently, SBV was silenced in *A. Cerana* by the application of exogenous dsRNA produced in *Bacillus thurigiensis* [21].

In our study, we initially performed a viral kinetics study to select the optimal viral dose and assess the progression of VLs over time. While we did not conduct a specific dose–response study for the dsRNA, our selection of the dsRNA dose was informed by previous research in the field. Based on this, the dsRNA dose used in the viral silencing experiment was set at 1 μg per bee [10,11].

At the beginning of the ABPV silencing experiment and up to day 2, all treatments exhibited similar behavior, with no noticeable increase in VLs. A slight increase in basal VLs was observed across all treatments, consistent with findings by Dalmon et al. (2019) who reported similar patterns of DWV VLs under varying temperature conditions [22]. This increase may be attributed to stressful factors, such as handling bees outside their hive, isolation from the colony, or the absence of the queen, which could act individually or synergistically to influence viral replication. This suggests that such external factors, though non-specific, can contribute to a uniform baseline rise in VLs across experimental groups. Significant differences were observed on days 4 (*p* = 0.0006) and 6 (*p* = 0.0002), with treatments C (ABPV 10^4^ gc/μL) and E (ABPV 10^4^ gc/μL + non-specific dsRNA) showing a peak in viral replication. These peaks suggest that ABPV replication was uninhibited in these groups, potentially due to the absence of effective RNA interference mechanisms. In contrast, treatment B (ABPV 10^4^ gc/μL + dsRNA) showed a pronounced suppression of VLs, indicating that specific dsRNA successfully triggered the silencing pathway, mitigating viral replication. This outcome highlights the efficacy of targeted RNA silencing in controlling ABPV, underscoring its potential as a biotechnological tool for managing viral infections in bees. The statistical significance of the differences (*p* = 0.0006 on day 4 and *p* = 0.0002 on day 6) further supports the reliability of these findings. On day 6, significant differences in average VLs were observed between treatments B and C. Crucially, the reduction in VLs in treatment B correlated with significantly higher bee survival rates. By day 5 (2 days post-infection), treatment B demonstrated a statistically significant increase in live bees compared to treatment E, which lacked specific RNA interference. By day 6 (3 dpi), survival in treatment B was significantly greater than in treatments C and E, further underscoring the protective effects of specific dsRNA. These results highlight the dual benefits of targeted RNA silencing: reducing viral replication and enhancing host survival (Table 3). These findings are consistent with Desai et al. (2012), who demonstrated that feeding DWV-specific dsRNA reduced the adverse effects of DWV infections [11].

Additionally, in this study, we estimated that dsRNA oral administration resulted in a decrease in average VLs from 2.48 to 3.48 log gc/μL by day 6 (3 dpi).

One of the most promising results was that treatment D (dsRNA) showed no toxic effects and performed similarly to controls. By day 8, it is likely that depletion or degradation of specific dsRNA contributed to the observed increase in viral load.

The purpose of this development is to apply RNAi technology to mitigate the impact of viral infections in queen-rearing apiaries and during the transportation of colonies or queens. These processes require a robust antiviral treatment with sustained efficacy over several days. In this study, we initially evaluated the antiviral effects of specific dsRNA over a 10-day period. However, our findings indicate that further work is necessary to ensure prolonged efficacy. Future studies could explore administering a second dsRNA dose on day 5, using higher dsRNA concentrations, or targeting additional genes to extend the antiviral effect.

This work demonstrated the significant potential of RNAi technology in controlling ABPV replication in honey bees, as evidenced by its ability to reduce viral loads and enhance survival in controlled experiments. These results also highlight the need to evaluate its efficacy against other viral agents affecting honey bee health and to explore the scalability of dsRNA-based treatments for practical application in beekeeping.

Although scalability is feasible, it presents several challenges and constraints that must be addressed. Application of RNAi technology in field settings requires overcoming barriers related to production, delivery, and cost. Large-scale synthesis of dsRNA has become more efficient with advances in biotechnological methods; however, cost-effectiveness remains a key concern for widespread adoption in beekeeping practices.

Delivery methods also require optimization to ensure efficacy under natural conditions. Potential approaches include incorporating dsRNA into sugar syrup, pollen patties, or sprays that can be readily consumed by bees. Ensuring stability and protection of dsRNA from environmental degradation, such as UV exposure or enzymatic breakdown, is critical for its success. Encapsulation technologies, such as nanoparticle-based delivery systems, have shown promise in enhancing dsRNA stability and targeted delivery [23].

Production of dsRNA at a cost feasible for field application remains a critical issue. Certain approaches have been evaluated, such as microbial production using genetically engineered bacteria like *Escherichia coli* [15,24]. This system is scalable, cost-effective, and environmentally sustainable, making it suitable for agricultural use. Another promising technique involves yeast-based systems [25] or plant-based production [26], which could further reduce costs.

Overall, the data indicate that specific dsRNA treatment (Treatment B) provided significant protective effects against ABPV infection, resulting in higher survival rates compared to the virus-only and non-specific dsRNA groups. This suggests that RNA interference mediated by specific dsRNA effectively reduces viral load and improves bee survival. The decreased survival observed in treatments C and E highlights the virulence of ABPV and the ineffectiveness of non-specific dsRNA.

These findings underscore the potential of targeted dsRNA treatments as a viable strategy for mitigating viral infections in honey bee populations. Further research is needed to optimize dsRNA delivery methods and to assess the long-term impacts on colony health and productivity.

## 5. Conclusions

For the first time, ABPV-dsRNA targeting just two genes significantly reduces the mortality caused by ABPV infection. This dsRNA can be used to prevent viral infections during the critical periods of queen production. Additionally, administrating dsRNA prior to queen transport for export, typically a stressful process associated with increasing mortality, could help mitigate this risk by reducing baseline virus levels. Furthermore, dsRNA could be employed to produce virus-free hives for laboratory experiments or to inhibit viral replication in vitro cultures where controlling baseline virus presence is a major challenge. Continued research into virus prevention techniques is essential to minimize losses in queen-rearing production and ensure the highest health standards for marketable products.

## Figures and Tables

**Figure 1 insects-16-00453-f001:**
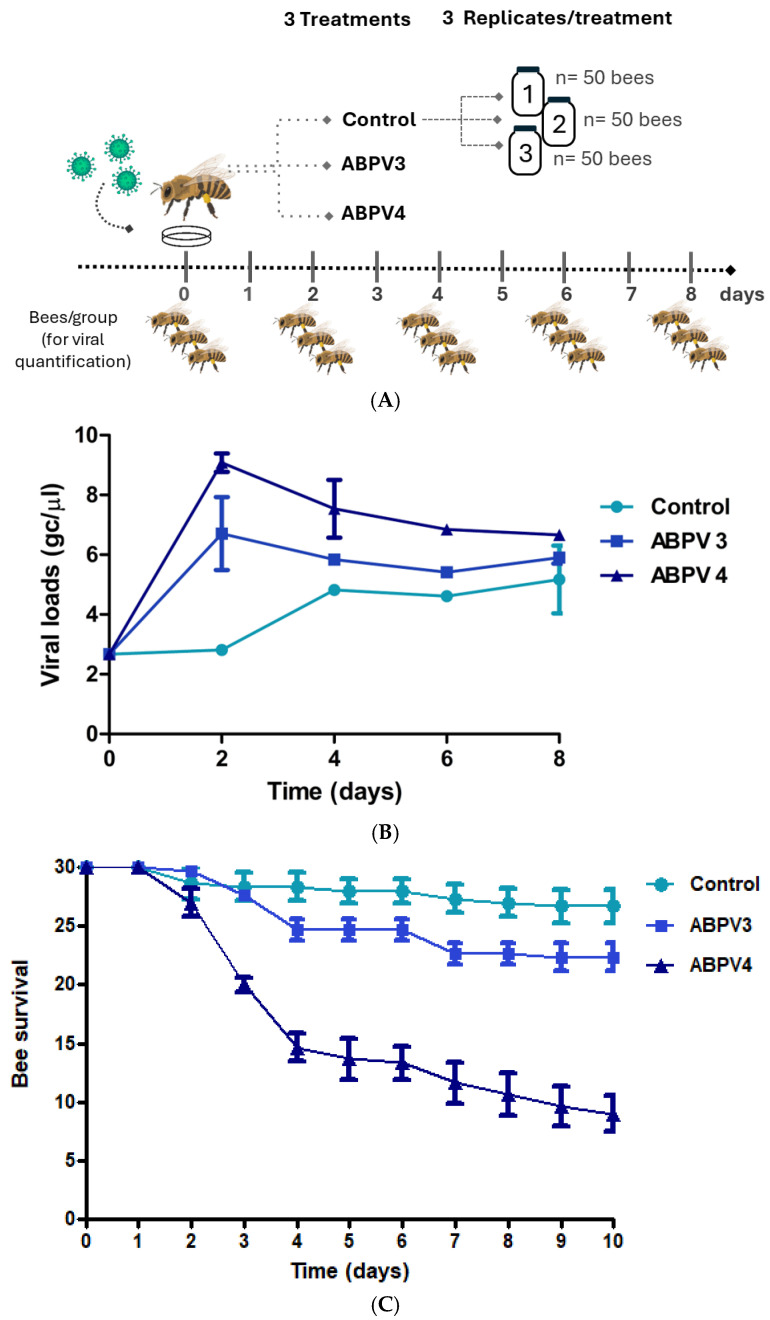
ABPV infection in adult bees. (**A**) Graphic representation of the experimental design. The treatments were: ABPV4: 10^4^ cg/μL; ABPV3: 10^3^ cg/μL; Control: without virus. (**B**) Viral loads (log genome copies/μL;) over time for each treatment (triangles: ABPV4, 10^4^ gc/μL; squares: ABPV3, 10^3^ gc/μL, and circles: control without virus). (**C**) Analysis of bee survival rate based on the number of live bees per glass jar across treatments and days of the experiment (triangles: ABPV4, 10^4^ cg/μL; squares: ABPV3, 10^3^ cg/μL and circles: control without virus).

**Figure 2 insects-16-00453-f002:**
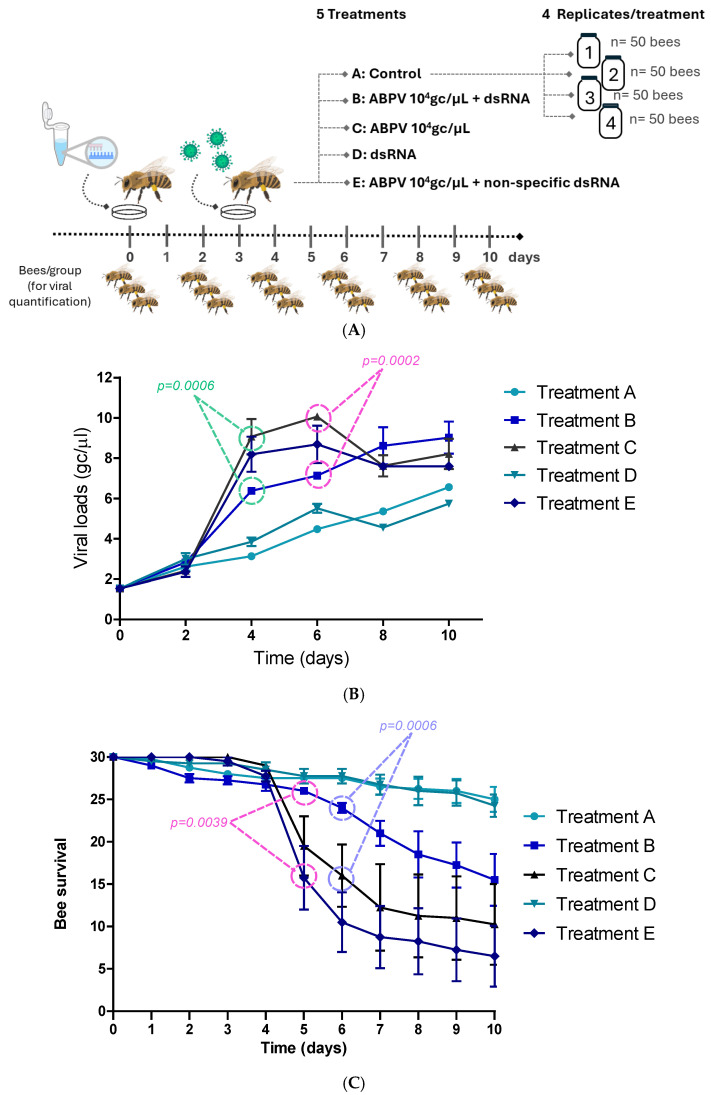
ABPV silencing experiment. (**A**) Graphic representation of experimental design. (**B**) Viral loads (log gene copies/μL) over time for each treatment. Treatment A: Control (circles); Treatment B: ABPV 10^4^ gc/μL + dsRNA (squares); Treatment C: ABPV 10^4^ gc/μL (black triangles); Treatment D: dsRNA (light blue triangles); Treatment E: ABPV 10^4^ gc/μL + non-specific dsRNA (rhombus). (**C**) Analysis of bee survival rate based on the number of live bees per glass jar across treatments and days of the experiment. Treatment A: Control (circles); Treatment B: ABPV 10^4^ gc/μL + dsRNA (squares); Treatment C: ABPV 10^4^ gc /μL (black triangles); Treatment D: dsRNA (light blue triangles); Treatment E: ABPV 10^4^ gc/μL non-specific dsRNA (rhombus).

**Table 1 insects-16-00453-t001:** Description of the treatments and groups used in the infection experiment: Treatments ABPV3 and ABPV4 involved the oral administration of honey containing the viral inoculum, whereas the control treatment consisted of feeding bees only honey.

Groups.	Treatment	Description
1-2-3	Control	Without virus
4-5-6	ABPV3	ABPV 10^3^ gc/µL
7-8-9	ABPV4	ABPV 10^4^ gc/µL

**Table 2 insects-16-00453-t002:** Description of the treatments and groups used in the silencing experiment: Treatments B–E involved the oral administration of honey containing the viral inoculum (C), the viral inoculum and 50 µg dsRNA (B, E) or 50 µg dsRNA (D). Control treatment (A) consisted of feeding bees only honey.

Groups.	Treatment	Description
1-2-3-4	A	Control
5-6-7-8	B	ABPV 10^4^ gc/µL + dsRNA
9-10-11-12	C	ABPV 10^4^ gc/µL
13-14-15-16	D	dsRNA
17-18-19-20	E	ABPV 10^4^ gc/µL + dsRNA non-specific

**Table 3 insects-16-00453-t003:** Contrasts between treatments from day 5 onwards, along with corresponding *p*-values, were obtained by fitting a generalized linear mixed model and comparing treatments using orthogonal contrasts.

Day	Contrasts	*p*-Value
5	Treatment B—Treatment A	0.712
	Treatment B—Treatment C	0.0757
	Treatment B—Treatment D	0.6693
	Treatment B—Treatment E	0.0039
6	Treatment B—Treatment A	0.3841
	Treatment B—Treatment C	0.0195
	Treatment B—Treatment D	0.3532
	Treatment B—Treatment E	<0.0001
7	Treatment B—Treatment A	0.1527
	Treatment B—Treatment C	0.0048
	Treatment B—Treatment D	0.1365
	Treatment B—Treatment E	<0.0001
8	Treatment B—Treatment A	0.0373
	Treatment B—Treatment C	0.0126
	Treatment B—Treatment D	0.0432
	Treatment B—Treatment E	0.0002
9	Treatment B—Treatment A	0.0166
	Treatment B—Treatment C	0.0263
	Treatment B—Treatment D	0.0195
	Treatment B—Treatment E	0.0002
10	Treatment B—Treatment A	0.0069
	Treatment B—Treatment C	0.0487
	Treatment B—Treatment D	0.0119
	Treatment B—Treatment E	0.0003

## Data Availability

The original contributions presented in this study are included in the article/Appendix A. Further inquiries can be directed to the corresponding author.

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
