# Peer review of "Reduction in Acute Bee Paralysis Virus Infection and Mortality in Honey Bees (Apis mellifera) by RNA Interference Technology"

_insects, 2025, doi:10.3390/insects16050453_

Round 1
Reviewer 1 Report
Comments and Suggestions for Authors
The authors give a good work for this field. Acute bee paralysis was originally discovered as an inapparent infection of honeybees in Britain (Bailey etal., 1963). This paper attempts to use RNAi to reduce the replication of ABPV in bees. The design of the experiment is clear and reasonable. But the results are a bit difficult to understand and the discussion is not sufficient. The introduction section needs to further explain the symptoms and pathology of bees infected by ABPV. The results section showed that the virus load of ABPV in control group bees gradually increased with time, while the treatment group inoculated with the virus showed a trend of first increasing and then decreasing. By the 8th day, the virus load of the three treatments is close (Figure 1), the similar trend showed in Figure 2. That is a bit of strange, why the virus keep constantly increase at a low load amount at initiation (control group), but a different trend for ingestion a high dose of IAPV (group B and C in figure 1, group B and E in figure 2). Explanation is needed for the result. And it may take a longer time to observe the trend of virus load in each group for the test last only for 10 days. Other suggestions list as follow.
1. L90-95, The primer and gene information for specific dsRNA and non-specific should be given clearly.
2. L167-168, It is hard to understand how much of ABPV was ingested for each of honey bee? From the description, should I think that in treatment B and C there was at least 50ml ABPV put in the jar for the honey bees ingesting uncontrolled, so the amount of ABPV for each bees is different, or did you feed the bees one by one by any tools constantly or at one-time. The description should be clearly.
3. L184-185, Treatment B, E, ABPV and dsRNA were mixed before providing to the bees, or one by one and how? DsRNA was diluted in honey or sucrose for feeding to the bees?
4. L214, 104 gc/µL should be 104 gc/µL.
5. L271 cg/µL should be gc/µL.
6. Supplementary figure 1, The molecular weight marker in the graph should be Clearly labeled the size of each line in the marker.
Author Response
Thank you very much for taking the time to review this manuscript. Please find the detailed responses (in blue) below and the corresponding revisions/corrections highlighted/in track changes in the re-submitted files
REVIEWER 1
The authors give a good work for this field. Acute bee paralysis was originally discovered as an inapparent infection of honeybees in Britain (Bailey etal., 1963). This paper attempts to use RNAi to reduce the replication of ABPV in bees. The design of the experiment is clear and reasonable. But the results are a bit difficult to understand and the discussion is not sufficient.
The introduction section needs to further explain the symptoms and pathology of bees infected by ABPV.
Thank you for pointing this out. A brief explanation of the symptoms caused by Acute Bee Paralysis Virus (ABPV), was included.
The results section showed that the virus load of ABPV in control group bees gradually increased with time, while the treatment group inoculated with the virus showed a trend of first increasing and then decreasing. By the 8th day, the virus load of the three treatments is close (Figure 1), the similar trend showed in Figure 2. That is a bit of strange, why the virus keep constantly increase at a low load amount at initiation (control group), but a different trend for ingestion a high dose of IAPV (group B and C in figure 1, group B and E in figure 2). Explanation is needed for the result. And it may take a longer time to observe the trend of virus load in each group for the test last only for 10 days.
We agree with this comment. The results show that high viral loads ingested by the groups that received the virus or virus + non-specific dsRNA caused a peak in viral load (VL) one day post-infection, after which the VLs decreased and remained constant until the end of the experiment. This infection peak was avoided in the group that received virus + specific dsRNA, which also showed lower VLs. This suggests that the dsRNA acts efficiently during the early stages of infection.
In contrast, in the uninfected controls, only an increase in baseline viral loads was observed. The baseline VLs are significantly lower than the post-infection values, and their gradual increase is likely due to the stress generated during the experiment (handling, separation from the hive, absence of the queen, etc.)
Additionally, it is important to consider the peak of viral replication, as this factor is associated with mortality which is significantly different between treatments C and E compared to A and B in the silencing experiment. Therefore, viral loads should not only be considered on the last day of infection but also throughout the entire experiment. It is crucial to analyze the effects that the viral replication curve has on the survival of the bees.
Other suggestions list as follow.
- L90-95, The primer and gene information for specific dsRNA and non-specific should be given clearly.
A description of the non-specific dsRNA has been added to the Materials and Methods section. Additionally, the sequence of the primers used to amplify target genes and their corresponding amplicon sizes has been included in the manuscript.
- L167-168, It is hard to understand how much of ABPV was ingested for each of honey bee? From the description, should I think that in treatment B and C there was at least 50ml ABPV put in the jar for the honey bees ingesting uncontrolled, so the amount of ABPV for each bees is different, or did you feed the bees one by one by any tools constantly or at one-time. The description should be clearly.
The feeding schedule was clarified in the Materials and Methods section.
ABPV infection assay: “On day 0, bees assigned to treatments B and C were fed ad libitum with honey containing 1 ml of ABPV (at a concentration of 103 gc/μL and 104 gc/μL, respectively) per glass jar. Bees in treatment A (the control group), received only honey (table 1)”.
Viral silencing assay: “On day 0, bees in treatment A received only honey (control group). Bees in treatment B received specific dsRNA (at day 0) and virus (at day 3), treatment C received virus at day 3, treatment D received at day 0 specific dsRNA only, and treatment E received non-specific dsRNA (at day 0) + virus (at day 3) (table 2).
Each replicate (glass jar) was fed ad libitum with honey containing 1 ml of virus suspension containing 104 genomic copies/μl (2x105 gc/bee), and the dsRNA dosage per glass jar was 50 μg. The experiment lasted 10 days.”
Since the experimental unit is the glass jar and not each individual bee, ad libitum feeding has been provided, following the methodology used by the researcher Carrillo-Trip et al (2016).
Carrillo-Tripp, J. et al. In vivo and in vitro infection dynamics of honey bee viruses. Sci. Rep. 6, 22265; doi: 10.1038/srep22265 (2016).
- L184-185, Treatment B, E, ABPV and dsRNA were mixed before providing to the bees, or one by one and how? DsRNA was diluted in honey or sucrose for feeding to the bees?
The feeding schedule for the viral silencing assay was also clarified in the Materials and Methods section.
“On day 0, bees in treatment A received only honey (control group). Bees in treatment B received specific dsRNA (at day 0) and virus (at day 3), treatment C received virus at day 3, treatment D received at day 0 specific dsRNA only, and treatment E received non-specific dsRNA (at day 0) + virus (at day 3) (table 2)”.
ABPV and dsRNA were provided on different days, both diluted in honey.
- L214, 104 gc/µL should be 104gc/µL.
Corrected
- L271 cg/µL should be gc/µL.
Corrected
- Supplementary figure 1, The molecular weight marker in the graph should be Clearly labeled the size of each line in the marker.
The MWM was added to the figure for a better interpretation of the result.
Reviewer 2 Report
Comments and Suggestions for Authors
The authors should include a thorough explanation in the manuscript as to why they studied 10 days (The interference efficiency of RNAi generally lasts only about 3 days).
The authors should test the bee surviral rate, not number of bees alive.
Why Viral loads was continue to raise in the non-virus-infected groups? please explain. Virus in the process of proliferation can produce negative strand, virus titer test why not detect negative strand? The Discussion part of the manuscript should be significantly expanded and deepenedIn fact, RNAi is a very mature technology, and authors should compare the
infection of different fragments or different methods to improve the efficiency of
RNAi.
Author Response
Thank you very much for taking the time to review this manuscript. Please find the detailed responses below (in blue) and the corresponding revisions/corrections highlighted/in track changes in the re-submitted files.
REVIEWER 2
The authors should include a thorough explanation in the manuscript as to why they studied 10 days (The interference efficiency of RNAi generally lasts only about 3 days).
We conducted the experiment over 10 days, as our goal is to implement this technology in queen bee rearing, a process that lasts approximately 10 days depending on hive management. It would also be important to use it during the transportation of live material. The importance of using dsRNA in the queen bee rearing process is highlighted in the conclusions.
The authors should test the bee surviral rate, not number of bees alive.
Thank you for your suggestion. In fact, we have assessed a rate based on the number of live bees per jar. We appreciate your comment, and we have included this clarification in both the methodology and the graphs
Why Viral loads was continue to raise in the non-virus-infected groups? please explain.
The results show that high viral loads ingested by the groups that received the virus or virus + non-specific dsRNA caused a peak in viral load (VL) one day post-infection, after which the VLs decreased and remained constant until the end of the experiment. This infection peak was avoided in the group that received virus + specific dsRNA, which also showed lower VLs. This suggests that the dsRNA acts efficiently during the early stages of infection.
In contrast, the uninfected control group exhibited only an increase in baseline viral loads. These baseline VLs were significantly lower than the post-infection values observed in infected treatments. The gradual increase in baseline VLs is likely attributable to stress factors associated with the experimental conditions, such as handling, separation from the hive, and the absence of the queen. Although the colonies selected for the bioassays were healthy and showed no signs of viral infection, background virus at very low titers was present.
Virus in the process of proliferation can produce negative strand, virus titer test why not detect negative strand?
“The detection of viral replication is crucial for differentiating between an active infection and just the presence of virus particles in a host.
Evidence that a virus is actively infecting a host includes the presence of viral particles and structures within host cells, revealed by electron microscopy, preferably including a specific nucleic acid or serological probe to positively identify the virus. Another approach is to detect the non-structural proteins involved in virus replication, which for most positive-stranded RNA viruses are only produced after invasion and mark the start of a replication cycle. The specific detection of viral negative strand RNAs can serve as a marker of active replication of these RNA viruses in a certain host, tissue or cell type”
COLOSS BEEBOOK VOL II. Standard methods for Apis mellifera pest and pathogen research. https://www.tandfonline.com/doi/pdf/10.3896/IBRA.1.52.4.22?needAccess=true
Detection of the negative strand is typically conducted to assess viral replication. However, in our experiment, we considered that it was unnecessary to specifically confirm ABPV replication in adult bees.
During infection, both negative and positive RNA strands are produced. The positive strand is associated with virions generated during viral proliferation. Increasing viral loads, as determined by detecting the positive strand of the VP2 gene, are indicative of active viral replication in the host. This approach is widely used to evaluate the presence of bee viruses, including ABPV.
On the other hand, procedures to detect the negative strand are significantly more complex, and calculating viral loads from such assays is considerably more challenging.
The Discussion part of the manuscript should be significantly expanded and deepened
In response to the reviewer's suggestion, the Discussion section was further elaborated and expanded.
In fact, RNAi is a very mature technology, and authors should compare the infection of different fragments or different methods to improve the efficiency of RNAi.
The reviewer rightly highlights that RNAi technology is extensively utilized as a reverse genetic tool for functional genomic studies across various fields. The success of externally induced dsRNA-mediated gene silencing, however, depends on multiple factors, including the method of dsRNA administration, dosage, dsRNA length, and the specific target gene and tissue. Previous studies have demonstrated that dsRNA administered through feeding is an effective and practical approach for reducing viral loads in both larval and adult bees (Brutscher et al., 2015). Similarly, a recent study in insects reported an effective RNAi response by targeting multiple genes via co-treatment with dsRNAs (Shariff et al., 2022). In our study, we chose to evaluate multi-target dsRNAs as an innovative strategy for controlling bee viruses. Nevertheless, a comment regarding this issue was incorporated in the discussion.
Brutscher LM, Flenniken ML. RNAi and Antiviral Defense in the Honey Bee. J Immunol Res. 2015;2015:941897. doi: 10.1155/2015/941897. Epub 2015 Dec 21. PMID: 26798663; PMCID: PMC4698999.
Sharif, M.N., Iqbal, M.S., Alam, R. et al. Silencing of multiple target genes via ingestion of dsRNA and PMRi affects development and survival in Helicoverpa armigera. Sci Rep 12, 10405 (2022). https://doi.org/10.1038/s41598-022-14667-z
Reviewer 3 Report
Comments and Suggestions for Authors
This manuscript describes a successful use of dsRNA for viral control in honey bees. It is an interesting analysis and will be of value to the scientific community. I recommend it be published after revision. My comments are below. Please note that it should be “honey bee” rather than “honeybee” and remove the green highlighting on the keywords in the front page.
Abstract & Introduction. I thought the abstract and introduction were good. My only suggestion would be to better describe the impact or effects of APBV on beekeeping in Argentina and around the world. Currently all that is noted is that it is present (lines 68-69).
Materials & methods, Lines 90-95. I wonder why the genome of APBV wasn’t used or cited for the development of dsRNA here? Should it be cited? The Teixeira et al. (2008) name is misspelt on line 91 and there is an errant period on that line too. Can the authors please provide more information on the dsRNA positive control sequence on line 100.
Materials & methods, lines 90-138. The primers used in the study for viral identification and quantification need to be cited somewhere, even if in the supplementary material.
Results, Lines 220-222. A little more information would be useful here on how long it took bees to develop the clinical signs of the ABPV infection would be useful here.
Results, tables. I think each of the table headings could benefit from more information, including the concentration of dsRNA used. Table 2 is not a description of the treatments and groups, but shows statistical tests. Given there are so many tests, a Bonferroni adjustment would be appropriate.
Results, Figure 2. Rather than a key with “treatment A”, etc, I’d suggest the authors use the specific treatment (ABPV + dsRNA, etc). Use periods rather than commas as decimal points.
Discussion, Lines 318-319. The authors state that “Additionally, in this study, we estimated that each μg of dsRNA administered re-sulted in a decrease in average VLs from 2.48 to 3.48 log gc/μL by day 6 (3 dpi)”. Given the authors had so few concentrations, could this really be said?
Discussion, Lines 320-321. The authors say that “One of the most promising results was that treatment D (dsRNA) showed no toxiceffects and performed similarly to controls”. Is that true? The ABPV plus non-specific dsRNA had the highest mortality in Figure 2c.
Conclusions. I suggest that the authors re-emphasize the importance of APBV here. Would it really be worthwhile to produce and use it? More reference to other work looking at dsRNA for viral control in bees might be useful somewhere in the discussion. Concentrations, used, etc.
References. Please check that your references are cited as the journal format.
Author Response
Thank you very much for taking the time to review this manuscript. Please find the detailed responses below (in blue) and the corresponding revisions/corrections highlighted/in track changes in the re-submitted files
REVIEWER 3
This manuscript describes a successful use of dsRNA for viral control in honey bees. It is an interesting analysis and will be of value to the scientific community. I recommend it be published after revision. My comments are below.
Please note that it should be “honey bee” rather than “honeybee” and remove the green highlighting on the keywords in the front page.
Thank you for noticing those mistakes. They were corrected.
Abstract & Introduction. I thought the abstract and introduction were good. My only suggestion would be to better describe the impact or effects of APBV on beekeeping in Argentina and around the world. Currently all that is noted is that it is present (lines 68-69).
Thank you for pointing this out. A brief explanation of the symptoms caused by Acute Bee Paralysis Virus (ABPV) and its impact was included.
Materials & methods, Lines 90-95. I wonder why the genome of APBV wasn’t used or cited for the development of dsRNA here? Should it be cited?
We agreed with this comment, and we have accordingly included the genome accession number in the Materials and Methods section
The Teixeira et al. (2008) name is misspelt on line 91 and there is an errant period on that line too.
Corrected
Can the authors please provide more information on the dsRNA positive control sequence on line 100.
Materials & methods, lines 90-138. The primers used in the study for viral identification and quantification need to be cited somewhere, even if in the supplementary material.
A description of the non-specific dsRNA has been added to the Materials and Methods section. Additionally, the sequence of the primers used for viral detection has been included in the Supplementary material.
Results, Lines 220-222. A little more information would be useful here on how long it took bees to develop the clinical signs of the ABPV infection would be useful here.
This is an interesting issue to address. The time it takes for honeybees to develop clinical signs of Acute Bee Paralysis Virus (ABPV) infection can vary based on several factors, including the viral load, environmental conditions, and the bee's overall health. Typically, clinical signs such as trembling, paralysis, and inability to fly manifest within a few days post-infection. These symptoms often escalate rapidly, leading to mortality in severe cases. Studies on ABPV-related pathology emphasize the role of overt infections triggered by stressors like environmental changes or co-infections, which can accelerate symptom development (Muz & Muz, 2023). However, there is variability, as some infected colonies may harbor the virus without immediate clinical signs, influenced by factors like nutritional status and genetic resistance (Nogueira et al., 2024).
De Miranda et al. (2010) reported that ABPV, KBV, and IAPV are highly virulent, requiring fewer than 100 particles to cause death within a few days when injected into pupae or adult bees. These viruses can rapidly kill their hosts, with mortality typically occurring 3–5 days post-inoculation with sufficient virion loads (COLOSS BEEBOOK Vol II).
However, there is a notable gap in studies that define the detailed kinetics of ABPV infection, such as the time to onset of clinical signs, peak viral titers, and viral clearance. In our study, we observed that viral loads peaked at 2 days post-infection (using an inoculum of 10⁴ gc ABPV/µl), coinciding with the onset of clinical signs. This was followed by a marked increase in mortality starting at day 4 post-infection.
Due to the limited information available on this subject, we have primarily focused on the results obtained in this study. However, if the reviewer considers it relevant, we would be glad to include additional comments addressing these findings and their broader implications.
Dilek, Muz., Mustafa, Necati, Muz. (2023). Acute bee paralysis virus field isolates from apiaries suffering colony losses in Türkiye. Journal of Apicultural Research, 62:1169-1175. doi: 10.1080/00218839.2023.2242123
M., F., Nogueira., Camila, Dantas, Malossi., M., Frediani., David, Pereira., Simone, S., Prado., João, Pessoa, Araújo., Cristiano, Menezes. (2024). Molecular Characterization of Acute Bee Paralysis Virus (ABPV) and Black Queen Cell Virus (BQCV) in Honeybees (Apis mellifera L. (Hymenoptera: Apidae)) from the Campinas Region. Sociobiology, 71(2):e10306-e10306. doi: 10.13102/sociobiology.v71i2.10306
de Miranda JR, Cordoni G, Budge G. The Acute bee paralysis virus-Kashmir bee virus-Israeli acute paralysis virus complex. J Invertebr Pathol. 2010 Jan;103 Suppl 1:S30-47. doi: 10.1016/j.jip.2009.06.014.
COLOSS BEEBOOK VOL II. Standard methods for Apis mellifera pest and pathogen research. https://www.tandfonline.com/doi/pdf/10.3896/IBRA.1.52.4.22?needAccess=true
Results, tables. I think each of the table headings could benefit from more information, including the concentration of dsRNA used. Table 2 is not a description of the treatments and groups, but shows statistical tests. Given there are so many tests, a Bonferroni adjustment would be appropriate.
Thank you for this useful suggestion. The table titles were expanded for better understanding of the results.
The table titles are located below them; table 2 describes the treatment and groups of the viral silencing assay and table 3 refers to the contrasts between treatments.
Orthogonal contrasts whit Bonferroni correction were performed to determine if there were significant differences between treatments at each time point. This was clarified in Materiales and Methods.
Results, Figure 2. Rather than a key with “treatment A”, etc, I’d suggest the authors use the specific treatment (ABPV + dsRNA, etc). Use periods rather than commas as decimal points.
Figure 2 was modified according to the reviewer suggestion.
Discussion, Lines 318-319. The authors state that “Additionally, in this study, we estimated that each μg of dsRNA administered re-sulted in a decrease in average VLs from 2.48 to 3.48 log gc/μL by day 6 (3 dpi)”. Given the authors had so few concentrations, could this really be said?
Thank you for your suggestion, we have reworded the paragraph in the manuscript. The reduction in viral loads observed is one of the key results of our experiment, reflecting the effect of dsRNA administration on the reduction of these viral loads. We believe it is important to highlight the impact achieved in our experiment, as it provides valuable information about the treatment's effectiveness.
Discussion, Lines 320-321. The authors say that “One of the most promising results was that treatment D (dsRNA) showed no toxic effects and performed similarly to controls”. Is that true? The ABPV plus non-specific dsRNA had the highest mortality in Figure 2c.
Comparison between treatment A (control) and treatment D (only dsRNA) demonstrated that dsRNA ingestion did not negatively impact bee survival.
Given that four replicates of treatment C were performed and the variations between them are minimal, the results are consistent. In contrast, for the replicates of treatment E, one of the jars shows high mortality, which lowers the group mean and significantly increases the variance. We thought that this high mortality could be attributed to the possible presence of other pathogens, intracellular parasites, or even an error in handling the jar. We did not investigate the cause of the variability observed in that jar and used the data as they were obtained.
Conclusions. I suggest that the authors re-emphasize the importance of APBV here. Would it really be worthwhile to produce and use it? More reference to other work looking at dsRNA for viral control in bees might be useful somewhere in the discussion. Concentrations, used, etc.
Thank you for pointing this out. The discussion was optimized according to the reviewer suggestions.
References. Please check that your references are cited as the journal format.
We have revised and corrected the Reference section.

Reviewer 4 Report
Comments and Suggestions for Authors
I support this kind of research because it is important in the field of bee health and helps combating viral infections. In this study for the first time, ABPV-dsRNA targeting ABPV specific genes was used with aim to reduce levels of ABPV in treated bees. Results revealed significant reduction in bee mortality caused by ABPV infection as well as decreasing of viral loads and mitigate risks connected with viral infections. This method has potential to be used in prevention of viral infections during the critical periods of beekeeping as well as honey bee queen production.
The title of the article is well-defined. The introduction is well written and contains all important information regarding this topic.
In the materials and methods section, some details are missing and needs to be improved. See comments bellow.
Results are well presented with all necessary data given through text and supported with adequate figures and tables. In the discussion authors corelated their findings with similar experiments and gave some hypothesis as well as ideas for further studies. The conclusions are clear and authors underlined the most important findings.
Some minor points:
Line 42: Please remove green colour in this line.
Line 42: Please consider adding “RNA interference technology” as keyword
Lines 91-92: the references were written two times. Please revise this.
Lines 171-172: It is not clear if you fed bees individually or you gave these 2 ml in the cage and bees were feed ad libitum.
Line 182: please explain what is (in this case) and how you produced non-specific RNA.
Lines 184-185: Again, how you infected them, individually, how you can be sure that each bee received exactly 1 microliter, add these details.
Line 186: “Three bees from each group were removed on days 0, 2, 4, 6 and 8, and” days post infection or from the beginning of the experiment.
Line 214: 104 - move 4 in superscript.
Line 221: In both infected groups? Were there any differences in signs in bees from group infected with higher amount of virus compared to other one?
Lines 236-237: “Significant differences were also observed on day 6 between treatments B and C (p=0.0002) (figure 2B).” Significant difference was in which way: higher or lower?
Line 313 and through the whole manuscript: “By day 5 (2 days post-infection)” it is not clear when was the infection. According to this, and similar statements 5-2=3 the infection was on day 3, however, in Figures 1 and 2 showing that infection was on day 1. Please revise this through the whole manuscript in text and in figures.
Author Response
Thank you very much for taking the time to review this manuscript. Please find the detailed responses below (in blue) and the corresponding revisions/corrections highlighted/in track changes in the re-submitted files
REVIEWER 4
I support this kind of research because it is important in the field of bee health and helps combating viral infections. In this study for the first time, ABPV-dsRNA targeting ABPV specific genes was used with aim to reduce levels of ABPV in treated bees. Results revealed significant reduction in bee mortality caused by ABPV infection as well as decreasing of viral loads and mitigate risks connected with viral infections. This method has potential to be used in prevention of viral infections during the critical periods of beekeeping as well as honey bee queen production.
The title of the article is well-defined. The introduction is well written and contains all important information regarding this topic.
In the materials and methods section, some details are missing and needs to be improved. See comments bellow.
Results are well presented with all necessary data given through text and supported with adequate figures and tables. In the discussion authors corelated their findings with similar experiments and gave some hypothesis as well as ideas for further studies. The conclusions are clear and authors underlined the most important findings.
Some minor points:
Line 42: Please remove green colour in this line.
Corrected
Line 42: Please consider adding “RNA interference technology” as keyword
The suggested keyword was added
Lines 91-92: the references were written two times. Please revise this.
Correction was done.
Lines 171-172: It is not clear if you fed bees individually or you gave these 2 ml in the cage and bees were feed ad libitum.
The feeding schedule was clarified in the Materials and Methods section.
ABPV infection assay: “On day 0, bees assigned to treatments B and C were fed ad libitum with honey containing 1 ml of ABPV (at a concentration of 103 gc/μL and 104 gc/μL, respectively) per glass jar. Bees in treatment A (the control group), received only honey (table 1)”.
Viral silencing assay: “On day 0, bees in treatment A received only honey (control group). Bees in treatment B received specific dsRNA (at day 0) and virus (at day 3), treatment C received virus at day 3, treatment D received at day 0 specific dsRNA only, and treatment E received non-specific dsRNA + virus (table 2). Each replicate (glass jar) was fed ad libitum with honey containing 1 ml of virus suspension containing 104 genomic copies/μl (2x105 gc/bee), and the dsRNA dosage per glass jar was 50 μg”.
Line 182: please explain what is (in this case) and how you produced non-specific RNA.
A description of the non-specific dsRNA has been added to the Materials and Methods section.
Lines 184-185: Again, how you infected them, individually, how you can be sure that each bee received exactly 1 microliter, add these details.
Since the experimental unit is the glass jar and not each individual bee, ad libitum feeding has been provided, following the methodology used by the researcher Carrillo-Trip et al (2016).
Carrillo-Tripp, J. et al. In vivo and in vitro infection dynamics of honey bee viruses. Sci. Rep. 6, 22265; doi: 10.1038/srep22265 (2016).
Line 186: “Three bees from each group were removed on days 0, 2, 4, 6 and 8, and” days post infection or from the beginning of the experiment.
Those days are calculated from the beginning of the experiment.
Line 214: 104 - move 4 in superscript.
Corrected
Line 221: In both infected groups? Were there any differences in signs in bees from group infected with higher amount of virus compared to other one?
The clinical signs observed in both groups were similar, including trembling, inability to fly, and gradual darkening of the body. However, a noticeable difference in mortality was evident between the groups.
Lines 236-237: “Significant differences were also observed on day 6 between treatments B and C (p=0.0002) (figure 2B).” Significant difference was in which way: higher or lower?
At that time point, treatment C showed significantly higher VL, in comparison to treatment B. The same was observed when mortality was evaluated.
Line 313 and through the whole manuscript: “By day 5 (2 days post-infection)” it is not clear when was the infection. According to this, and similar statements 5-2=3 the infection was on day 3, however, in Figures 1 and 2 showing that infection was on day 1. Please revise this through the whole manuscript in text and in figures
Thank you very much for noticing that mistake. Figure 2 was modified indicating correctly the days at which dsRNA and the viral inoculum were added. The whole manuscript was also revised for possible inconsistencies.
Round 2
Reviewer 2 Report
Comments and Suggestions for Authors
Dear Authors! Thanks for the corrections you’ve made. You have improved the text according to the comments.